# Encapsulation of Synthesized Plant Growth Regulator Based on Copper(II) Complex in Chitosan/Alginate Microcapsules

**DOI:** 10.3390/ijms22052663

**Published:** 2021-03-06

**Authors:** Darikha Kudasova, Botagoz Mutaliyeva, Kristina Vlahoviček-Kahlina, Slaven Jurić, Marijan Marijan, Svetlana V. Khalus, Alexander V. Prosyanik, Suzana Šegota, Nikola Španić, Marko Vinceković

**Affiliations:** 1Biotechnology Department, M. Auezov South-Kazakhstan University, Tauke-Khan av. 5, 160012 Shymkent, Kazakhstan; darihauko@mail.ru (D.K.); mbota@list.ru (B.M.); 2Department of Chemistry, Faculty of Agriculture, University of Zagreb, Svetošimunska 25, 10000 Zagreb, Croatia; kvkahlina@agr.hr (K.V.-K.); sjuric@agr.hr (S.J.); marijan.marijan1986@gmail.com (M.M.); 3Ukrainian State University of Chemical Technology, Gagarina av., 49000 Dnipro, Ukraine; swetasnegur9@gmail.com (S.V.K.); prosyanykav@gmail.com (A.V.P.); 4Laboratory for Biocolloids and Surface Chemistry, Ruđer Bošković Institute, 10000 Zagreb, Croatia; Suzana.Segota@irb.hr; 5Department of Wood Technology, Faculty of Forestry, University of Zagreb, 10000 Zagreb, Croatia; nspanic@sumfak.hr

**Keywords:** copper(II) complex, plant growth regulation, encapsulation, controlled release

## Abstract

A new copper complex, trans-diaqua-trans-bis [1-hydroxy-1,2-di (methoxycarbonyl) ethenato] copper (abbreviation Cu(II) complex), was synthesized and its plant growth regulation properties were investigated. The results show a sharp dependence of growth regulation activity of the Cu(II) complex on the type of culture and its concentration. New plant growth regulator accelerated the development of the corn root system (the increase in both length and weight) but showed a smaller effect on the development of the wheat and barley root systems. Stimulation of corn growth decreased with increasing Cu(II) complex concentration from 0.0001% to 0.01% (inhibition at high concentrations—0.01%). The development of corn stems was also accelerated but to a lesser extent. Chitosan-coated calcium alginate microcapsules suitable for delivery of Cu(II) complex to plants were prepared and characterized. Analysis of the FTIR spectrum showed that complex molecular interactions between functional groups of microcapsule constituents include mainly electrostatic interactions and hydrogen bonds. Microcapsules surface exhibits a soft granular surface structure with substructures consisting of abundant smaller particles with reduced surface roughness. Release profile analysis showed Fickian diffusion is the rate-controlling mechanism of Cu(II) complex releasing. The obtained results give new insights into the complexity of the interaction between the Cu(II) complex and microcapsule formulation constituents, which can be of great help in accelerating product development for the application in agriculture

## 1. Introduction

The plant growth regulators (PGRs) are organic compounds, either produced naturally within the plants (phytohormones) or synthesized in laboratories, that modify or control one or more specific physiological processes within a plant. They act on the plant physiology, stimulate, or slow down the rate of growth or maturation or otherwise change the behavior of plants or their products. Specific PGRs are used to modify crop growth rate and growth pattern during the various stages of development from germination through harvest and post-harvest preservation [1,2]. Phytohormones are produced in low concentrations in plants, and many studies are focused on the preparation of synthetic plant growth regulators (PGRs) suitable for plant growth and development in agricultural practice. The economic or agronomic benefits from the use of synthetic PGRs are many times higher than the costs made when studying the spectrum of the biological activity of natural PGRs. 

It is well-known that copper is one of the most important microelements with many functions in plants. It has an important key role in photosynthetic and respiratory electron transport chains, participates in many physiological processes, and behaves as a structural element in some metalloproteins involved in electron transport in chloroplasts and mitochondria, as well as in plant oxidative stress response, as a cofactor in enzymes such as Cu/Zn-superoxide dismutase (Cu/ZnSOD), cytochrome c oxidase, ascorbate oxidase, amino oxidase, laccase, plastocyanin, and polyphenol oxidase [3]. In the case when copper is not available or deficient, plants develop specific symptoms, most of which affect young leaves and reproductive organs causing problems in normal plant growth and development. Although copper is an essential element, it could be also a source of inherent toxicity and negative impact on plant growth and development. The higher excess of copper could cause disorders in plant growth and development by adversely affecting important physiological processes in plants [4]. Copper can initiate oxidative damage and interfere with the important cellular processes such as photosynthesis, pigment synthesis, plasma membrane permeability, and other metabolic mechanisms, causing a strong inhibition of plant development and growth, even leading to plant withering to death [5,6].

Besides many functions in plants, copper cations also possess fungicidal and bactericidal properties. Copper compounds, like copper sulfate, have been traditionally applied to soils deficient in copper, although sometimes the effects may be negative. There is a very high risk to use soluble CuSO_4_ because when applied in excess, soil copper levels become toxic to plants and can affect human health [7]. Due to all of the above, there is an urgent need to modify and improve commercial copper fertilizers. Current investigations are mainly focused on developing a new generation of enhanced agroformulations based on copper more efficiently than traditional fertilizers thus reducing environmental pollution [8,9,10,11]. Various chelate complexes are used in agricultural practice due to their biocidal activity toward some fungi and bacteria and as a supplier of essential microelements to the plants [12]. Some complexes with Cu(II) have been reported to be useful in agriculture as plant growth-regulators [13,14,15,16]. 

Another approach is to use encapsulated copper or copper compounds which allow the use of smaller amounts and controlled release [17]. Encapsulation is an advanced technology of inserting an active component into the polymer matrix to form particles (within micro- or nanoscale). The advantages of using microcapsules in agriculture are in protecting the encapsulated active agent from degradation (i.e., stabilization), reducing the amount required, and achieving controlled release to regulate growth for the entire crop cycle. This technology is recognized as an effective method for the controlled release of fertilizers [18,19,20].

We have synthesized trans-diaqua-trans-bis [1-hydroxy-1,2-di (methoxycarbonyl) ethenato] copper (abbreviation Cu(II) complex) and detected its plant growth-regulating activity. Any organic compound capable of influencing plant development is a plant growth regulator, including the above complex with the Cu(II) cation. As an organic component, the complex contains the enol form of oxaloacetic acid, which is part of the Krebs tricarboxylic acid cycle. In turn, in the Krebs cycle, almost all metabolic pathways for the conversion of carbohydrates and amino acids in all living organisms are realized. Accordingly, the complex under consideration has been created on a natural organic compound that plays an important role in the metabolism of substances in any biological cell. The considered complex was first obtained by the authors of the article.

The present study aims to design, prepare, and characterize microcapsules suitable for delivery of synthesized Cu(II) complex to the plants as a new agroformulation. The main hypothesis of this article is that by encapsulating the Cu(II) complex, which acts as a source of essential trace elements and a plant growth-regulator, we can prepare an improved agroformulation with copper delivery for the entire crop cycle. An additional advantage is that by encapsulating in biopolymer microcapsules formed with calcium as a gelling cation, we supply the plants with an additional important macroelement, calcium. The presence of micro-and macroelements in microcapsules which simultaneously affect the growth, protect and nourish the plant can provide wider possibilities for application on different plant cultures during their life cycle.

## 2. Results and Discussion

The results and discussion consist of three interrelated parts. The first part evaluates Cu(II) complex plant growth-regulating activity, the second part analyzes the molecular interactions between the microcapsule formulation constituents, and the third part discusses the important physicochemical properties of microcapsules loaded with Cu(II) complex.

### 2.1. Evaluation of Plant Growth-Regulating Activity of Novel Cu(II) Complex 

Results of seed-germination testing on the vegetative development and growth of individual crops are presented in Table 1. Three test concentrations (0.0001, 0.001, and 0.01%) were applied to three plant cultures (corn, barley, wheat). The results showed a sharp dependence of the Cu(II) complex influence on the type of culture. So, for corn, the increase in the growth-promoting properties of the tested Cu(II) complex with a decrease in their concentration from 0.01% to 0.0001% solution significantly exceeding the standard was observed. By applying Cu (II) complexes at a concentration of 0.0001%, the development of the corn root system reached ~1.5 times higher length and weight. The development of plant stems was also accelerated but to a lesser extent. This is important for their further growth, especially in adverse weather conditions (lack of moisture). Cu(II) complex exerts a significantly lesser effect on the development of the root system of wheat and barley. In the latter case, it is practically absent. Values of the deviation probabilities of the null hypothesis P, obtained from the data for Cu (II) in the concentrations 0.01, 0.001, and 0.0001% (only data with *p* < 0.05 were taken into account) are presented in Table 1.

Additional analysis of plant growth-regulating properties was performed using bioassays on auxin-like and gibberellin-like activity. The effect of the Cu(II) complex on the growth of the segment of corn coleoptiles in 24 h of exposure is presented in Table 2.

It can be seen that 1 cm pieces of corn coleoptiles during their incubation in the water grew by 2.3–3.5 mm. The impact of IAA stimulated their growth by 2.5 times to 7.3–7.9 mm. Such an increase in the lengths of coleoptiles in auxin is normal and confirms the high sensitivity of this test to auxin. The Cu (II) complex showed only a mild auxin-stimulating effect on the growth of corn coleoptile segments, which almost disappeared with increasing concentration.

The effect of Cu(II) complex at concentration 0.1 mg L^−1^ on the growth of whole corn coleoptiles are presented in Table 3a. It can be seen that Cu(II) complex slightly reduced leaf growth in 24 h of exposure (Table 3a). It turned out that corn, which has its auxin, reacts to its introduction from outside by increasing the gain by 33% compared to the control.

The results of experiments to identify the gibberellin-like activity of the Cu(II) complex are presented in Table 3b. It can be seen the gibberellic acid (GA) at a concentration of 10 mg L^−1^ accelerated the growth of the first leaf of coleoptiles almost 1.5 times, which indicates a high sensitivity of this test object to GA. In contrast, the Cu(II) complex used at a concentration of 0.1 mg L^−1^ only slightly reduced leaf growth in the 24 h of exposure, i.e., a biological test for gibberellin-like activity showed no stimulatory effect on the growth of 1st leaf seedling corn relative to control.

Testing of Cu(II) complex growth regulation activity shows a sharp dependence on the concentration and type of culture (corn, barley, wheat). Cu(II) complex accelerated the development of the corn root system (the increase in both the length and mass) and exhibited a significantly lesser effect on the development of the wheat and barley root systems. The development of corn plant stems was also accelerated but to a lesser extent.

L Biological examination of the Cu (II) complex on coleoptile segments showed no effect similar to auxin or gibberellin. The growth-promoting effects presented in Table 1 are due to the different mechanisms of action of the complex on plant development.

### 2.2. Molecular Interactions in Microcapsule Formulations

The FTIR spectra of dry microcapsule formulations prepared with or without Cu(II) complex, and of single constituents, Cu(II) complex, sodium alginate, chitosan, and calcium chloride are presented in Figure 1. The most characteristic bands of the Cu(II) complex are in correlation with the literature data of similar copper compounds [21,22,23]. Characteristics of Cu(II) complex spectra are peaks at 3433 cm^−1^ corresponding to the stretching vibration of -OH groups, at 1740 cm^−1^ to COO^-^ and 1266 cm^−1^ to CH_2_ stretching vibrations. Esters of carboxylic acids as well as of ethers containing the =C-O---R grouping have a prominent band in the 1270–1150 cm^−1^ region and a less intense band between 1120 and 1130 cm^−1^. The region between 960 and 850 cm^−1^ indicates hydrogen bonding of O-H groups. A peak at 540 cm^−1^ is attributed to the Cu-O bond with copper(II) ion.

Analysis of a single calcium chloride spectrum was previously reported [24]. The characteristic peaks in the calcium chloride spectrum are at 3494, 3396, 3214, 1646, and 663 cm^−1^. The frequency range 3214–3494 cm^−1^ and the medium intensity band at 1646 cm^−1^ represent the bending forms of hydroxyl groups. The medium intensity peak at 663 cm^−1^ represents the stretching of the Ca-O bond.

Analyses of sodium alginate and chitosan spectra were previously reported [12]. Characteristic vibration of the sodium alginate spectrum is strong and broad absorption band in the range 3000–3600 cm^−1^ (O–H group), stretching vibration at 2920 cm^−1^ (the aliphatic C–H group), the bands at 1595 and 1405 cm^−1^ (asymmetric and symmetric stretching peaks of carboxylate (–COO^-^) groups), weak broad stretching vibration at 1295 cm^−1^ (skeletal vibrations), and bands from 1081 to 1026 cm^−1^ (antisymmetric stretching (C-O-C)).

Chitosan spectrum exhibits the strong and broad absorption band around 3330 cm^−1^ (O-H and N-H stretching), stretching vibrations at 2925 cm^−1^ and 2875 cm^−1^ (asymmetric and symmetric modes of C-H), the absorption band at 1648 cm^−1^ (amide I band), the bending vibrations at 1582 cm^−1^ (N-H stretching of N-from amine and amide II), band at 1425 cm^−1^ (CH_2_ scissoring), the medium peak at 1373 cm^−1^ (symmetrical CH_3_ deformation), whereas vibrations in the 1190–920 cm^−1^ region represent C-N stretching vibrations and overlap of the vibrations from the carbohydrate ring.

In comparison with spectra of single alginate and chitosan, the spectrum of CS/(ALG/Ca) exhibits changes in the absorption bands of the amino groups, carboxyl groups, and amide bonds confirming a complex formation between two oppositely charged polyelectrolytes [25]. The disappearance of chitosan distinct peaks is probably due to the very low chitosan concentration compared with alginate. Characteristic peaks of alginate corresponding to carboxylate groups are less intense with shifts to lower and higher frequencies, respectively.

The spectra of CS/(ALG/Ca) and CS/(ALG/(Ca+Cu)) are very similar but they differ in the position and intensity of the main bands. The presence of Cu(II) complex in the alginate matrix causes the most significant changes in the alginate functional groups region, hydroxyl (OH), ether (COC), and carboxylate (COO^-^) showing shifting to the higher wavenumbers (3301, 1632, 1440, and 1085 cm^−1^). The band at 1085 cm^−1^ corresponding to C-O a stretching vibration is referred to as the guluronic unit (also known as carbohydrate region). The stability of calcium alginate is determined by the transmittance of the guluronic unit which has been cross-linked with the calcium ion [26,27].

The weaker intensity of band around 3400 cm^−1^ in CS/(ALG/(Ca+Cu)) spectrum revealed the attenuation of hydrogen bonding due to Cu(II) complex encapsulation. Similar changes can be seen in the range of asymmetric and symmetric COO^-^ stretching indicating electrostatic interactions. The wavenumbers around 3323 cm^−1^ indicated that calcium alginate is a hydrogel rich in -OH groups. Due to the high contents of -OH groups in calcium alginate, this hydroxyl stretching vibration involves the bonding between alginate molecules and also the alginate-water interaction [28]. Results of FTIR analysis revealed intermolecular interactions between all microcapsule constituents include mainly hydrogen bonds and electrostatic interactions.

### 2.3. Physicochemical Characterization of Microcapsules

#### 2.3.1. Morphological Characterization of Microcapsules

The analysis of morphology, size, and shape of the wet and dry microcapsules with and without Cu(II) complex was performed immediately after preparation by light microscopy and after drying to constant mass (approximately four weeks on air at room temperature) by scanning electron microscopy. Typical light microscopy microphotographs of wet and dry CS/(ALG/Ca) and CS/(ALG/(Ca+Cu)) microcapsules are presented in Figure 2.

The morphological analysis of the wet microcapsules carried out soon after the preparation showed rounded external surfaces. The surface of wet microcapsules appears rough with a large number of irregularities but without visible cracks. Both types of wet microcapsules are almost spherical or oval with the mean particle size around 2 mm (for (CS/(ALG/Ca) *d =* 2053 ± 114 µm; CS/(ALG/(Ca+Cu)) *d =* 2056 ± 226 µm) (Figure 2a,c).

The loading of the Cu(II) complex did not significantly change microcapsule size or shape. The spherical or oval shapes are conditioned by the concentration of calcium chloride (1%) during preparation [29]. CS/(ALG/Ca) microcapsules are white and CS/(ALG/Ca+Cu)) microcapsules are light blue due to Cu(II) complex presence inside the matrix.

All microcapsules contain almost 97% of water. Water lost during drying to constant mass on air caused changes in microcapsules morphology, shape, and size (Figure 2b,d). Upon drying the spherical shape was disrupted and the surface of both microcapsule types becomes wrinkled. The mean particle size of dried microcapsule was reduced to 845.15 ± 98.23 µm for CS/(ALG/Ca) and 811.32 ± 103.21 µm for CS/(ALG/(Ca+Cu)). Size reduction for more than 60% is associated with biopolymer strain-relaxation processes [25].

SEM microphotographs of both microcapsule types presented in Figure 3a,c show elliptical shapes. Enlarged surface images revealed a fibrous network structure with numerous pores (Figure 3b,d and Figure 4a,c). Loading of Cu(II) complex in CS/(ALG/Ca) caused the change in surface homogeneity and the appearance of locally aggregated nubs on the surface (marked by white arrows in Figure 3c).

Analysis by energy-dispersive X-ray spectroscopy (EDX) applied to the area nearest to the surface of microcapsules (Figure 4a,c) revealed that the detected elements correspond to the content of both microcapsules (Figure 4b,d). The detection of copper indicates a part of the Cu(II) complex localized on the microcapsule surface (Figure 4d).

The appearance of copper near the microcapsule surface could be explained by the difference in the affinity of alginate to calcium and copper cations and the difference in their size (Cu^2+^ has a smaller radius than Ca^2+^). Other relevant factors may include differences in the charge density and affinity of the atomic orbitals to form the coordination bond [30].

To get a deep insight into the surface topology revealed by optical and SEM microscopy, AFM analysis was performed using 2D- and 3D-topographic height and phase images, and cross-section surface profiles are taken along the white lines in images plotted in Figure 5.

AFM imaging was performed at different regions of each CS/(ALG/(Ca+Cu)) microcapsule to ensure the consistency of the obtained results. The surfaces of microcapsules showed elongated not spherical grains with shorted axis dimensions around *d* = 500 nm and the longer axis around *d* = 700 nm (see red arrows in Figure 5c). Each observed grain occupies substructures consisting of abundant smaller particles with the uniform size (*d* = 50 nm–60 nm, Figure 5c blue arrows), but in comparison with CS/(ALG/Ca) [25], the boundaries between them are rather diffused than sharp. The very soft surfaces of investigated microcapsules were smoother than those made of CS/(ALG/Ca) suggesting the more compact structure formation. Loading with Cu(II) complex reduces microcapsules surface roughness from *R*_a_ = 26 nm ± 3 nm (CS/(ALG/Ca)) [25] to *R*_a_ = 6.2 nm ± 0.7 nm (CS/(ALG/(Ca+Cu))).

#### 2.3.2. DSC Analysis of Microcapsule Formulations

DSC thermograms of Cu(II) complex and microcapsules with or without loaded Cu(II) are presented in Figure 6a,b. Cu(II) complex thermogram (Figure 6a) shows several endothermic phase transitions, specific values of which are given in Table 4. The first two phase transitions are described with the formation of endothermic peaks at 94.6 °C and 108.1 °C. The formation of a rather broad peak at 94.6 °C and the formation of peak shoulder could be an indication of the overlapping effect of the loss of free water and solid-solid transition of dimethyl fumarate, to a phase that melts at 108.1 °C [31]. Endothermic peaks at 141.4 °C and 161.8 °C are most probably associated with loosing of crystal water molecules. It was shown that a variety of organic and inorganic hydrates give more than one dehydration endotherm by DSC due to a specific binding state or location of water molecules in the crystal lattice [32]. The endothermic peak at 223.8 °C corresponds to the melting of the Cu(II) complex.

Both control microcapsules and loaded with Cu(II) complex revealed the formation of three distinctive peaks, specific values of which are given in Table 4. Endothermic peaks at around 80 °C of both samples are associated with the loss of water due to the evaporation of free water and water bound to the hydroxyl groups [33]. The formation of this peak indicates that despite the drying process some water was still present on the microcapsule surface [34]. Thermogram also shows that peaks are rather broad with onset temperatures below 50 °C and end temperatures at around 115 to 120 °C, which is the indication of electrostatic interactions between chitosan and alginate matrix [35]. The second endothermic peak at 195 °C for CS/(ALG/Ca) sample and 181 °C for CS/(ALG/Ca+Cu) sample corresponds to the melting of calcium alginate structure, but can also be attributed to the partial polymer decomposition. Complete polymer decomposition is confirmed with the formation of exothermic peaks at 278.2 °C and 276.0 °C for CS/(ALG/Ca) and CS/(ALG/Ca+Cu)) samples, respectively [36,37].

The presence of Cu(II) in CS/(ALG/(Ca+Cu)) microcapsules shifted the second endothermic and exothermic peaks to lower temperatures because of the intermolecular hydrogen bonding and electrostatic interactions. Peaks of Cu(II) complex are not visible when loaded into a microcapsule probably due to the overlapping of their endotherms or too low amount of Cu(II).

#### 2.3.3. Encapsulation Efficiency, Loading Capacity, and Swelling Degree

Encapsulation efficiency and loading capacity determination were performed to obtain information on the yield and the content of Cu(II) complex in CS/(ALG/(Ca+Cu)) microcapsules. Their values depend on the selected biopolymer material, sodium alginate, which needs cross-linking reactions (copper and calcium ions), and the effect of two formulation variables (i.e., the ratio between chitosan and alginate) [38]. The value of encapsulation efficiency was 57.6 ± 0.1 % and loading capacity 25.9 ± 1.1 mg g^−1^.

By dispersing in water, the microcapsules are rehydrated, water fills the pores on the surface causing swelling without erosion/disintegration [39]. The swelling capacity of the gel microcapsules is an important factor for evaluating the usefulness of the materials for the controlled release of bioactive agents. Swelling degree data are listed in Table 5.

A very high swelling degree of CS/(ALG/Ca) may be ascribed to the high swelling and water uptake capabilities of chitosan [40]. It can be seen that loading with Cu(II) complex significantly decreases microcapsules swelling. One of the main factors that determine alginate swelling is a concentration of gelling cations during preparation having a major effect on both, the kinetics of gelation and the characteristics of the gel formed [40,41,42]. Both cations, calcium, and copper are gelling cations and cooperatively interact with a block of L-guluronic acid groups forming stronger ionic crosslinks between different alginate chains and denser networks than in CS/(ALG/Ca). Swelling is limited by crosslinks [43] because the penetration of water into the denser network with high cross-linking density is slower.

### 2.4. In Vitro Copper Release Profiles from Microcapsule Formulations

Microcapsules are small containers surrounded by a wall that can control the release from them. When dispersed in an aqueous solution, water penetrates the pores among the polymer chains causing swelling and release of loaded agents. It was shown that the most important rate-controlling release mechanisms from hydrophilic microparticles are diffusion, swelling, and erosion [44].

The release profile of Cu(II) complex from CS/(ALG/(Ca+Cu)) with time presented in Figure 7 is characterized by rapid release in the first hour followed by a slower release up to 500 h.

The burst initial release may be explained by the presence of a portion of Cu(II) complex on the microcapsule surface (nubs on the microcapsule surface, Figure 3c). After the initial rapid release, the release of Cu(II) complex located in the alginate matrix became slower due to the layer of a polyelectrolyte complex between positively charged amino groups of chitosan and negatively charged carboxylic acid groups of alginate on the microcapsule surface [45].

To identify the kinetics and type of release mechanism, a frequently used power-law model [46] modified for burst effect was applied [47]. Different controlling mechanisms may be distinguished by an empirical equation:(1)$$f=a+\;kt^{n}$$
where *a* is the y-axis intercept characterizing the burst effect, *k* is a kinetic constant characteristic for a particular system considering structural and geometrical aspects (a measure of the release rate), *n* is the release exponent representing the release mechanism, and *t* is the release time.

The magnitude of the release exponent *n* gives information about the release mechanism; *n* ≤ 0.45 characterizes Fickian diffusion, *n* ≥ 1.0 polymer relaxation/dissolution (type II transport), and *n* < 0.45 and *n* < 1.0 anomalous transport. Values of *n* between 0.45 and 1.0 can be regarded as an indicator of both phenomena (diffusion in the hydrated matrix and the polymer swelling and relaxation).

The values of *a,* the constant *k,* exponent *n,* and correlation coefficients of Cu(II) complex release from CS/(ALG/(Ca + Cu)) are listed in Table 6. The correlation coefficient (*R*^2^) was rather high indicating a good correlation within experimental data and Equation (4). *n* value below 0.45 indicates a classical Fickian diffusion-controlled release.

Based on the physicochemical characteristics and release mechanism prepared CS/(ALG/(Ca+Cu)) microcapsules could be used as an enhanced fertilizer for plant nutrition, protection, and plant growth promotor.

## 3. Materials and Methods

### 3.1. Materials

Low viscosity sodium alginate (CAS Number: 9005-38-3; Brookfield viscosity 4–12 cps (1% in H_2_O at 25 °C) was purchased from Sigma Aldrich (USA). Low molecular weight chitosan (CAS RN: 9012-76-4, molecular weight: 100,000–300,000 Da) was obtained from Acros Organic (USA). A commercially available product calcium chloride, CaCl_2_ was used as a calcium donating substance (Kemika Croatia). All other chemicals were of analytical grade and used as-received without further purification. Trans-diaqua-trans-bis [1-hydroxy-1,2-di (methoxycarbonyl) ethenato] copper was synthesized in the laboratory of Professor Alexander Prosyanik (Ukraine).

### 3.2. Synthesis of Trans-Diaqua-Trans-bis [1-Hydroxy-1,2-Di (Methoxycarbonyl) Ethenato] Copper

A solution of 0.85 g (0.005 mol) of CuCl_2_·4H_2_O in 10 mL of water was added to a solution of 1.59 g (0.01 mol) of aminofumaric acid dimethyl ester in 10 mL of isopropanol. After two days, the light green precipitate was separated, washed with 5 mL of a mixture of iso-propanol-water 1:1, crystallized, and dried.

Basic properties: yield 1.64 g (80%), T_melt._ 216–218 °C, molecular weight 417.8 g/mol; a light green crystalline substance with limiting solubility in water, alcohol, DMFA, and DMSO, insoluble in benzene, chloroform, carbon tetrachloride, ethers, and hexane.

### 3.3. Preparation of (CS/(ALG/(Ca + Cu)) Microcapsule Formulations

The preparation of microcapsule formulations was carried out in a two-step process, by the ionic gelation and polyelectrolyte complexation at ambient temperature as described by Vinceković et al. (2016) [12]. In brief, the Cu(II) complex was dissolved into 100 mL of sodium alginate solution (1.5 %) and homogenized by slight mixing and stirred (magnetic stirrer) for 60 min. The mixture (contains 1.5 % of Cu(II) complex) was dripped at the 30–40 mL min^−1^ flow into 100 mL of 1 mol dm^−3^ CaCl_2_ solution (contains 1.5% of Cu(II) complex) through the encapsulator nozzle size of 1000 μm at 40 Hz vibration frequency and 20 mbar pressure (Encapsulator Büchi- B390, Bütchi Labortechnik AG, Switzerland) under constant magnetic stirring. To promote gel strengthening, formed microspheres (ALG/(Ca+Cu)) was kept at room temperature for an additional 30 min. Microspheres were washed three times with sterile distilled water and filtered through the Büchner funnel.

In the second stage, washed microspheres were dispersed in 50 mL chitosan solution (0.5% CS in 1.0% CH_3_COOH) under constant stirring (magnetic stirrer). The contact time between microspheres and chitosan solution was about 30 min to give chitosan time to form a layer around the microspheres and formation of microcapsules, CS/(ALG/(Ca+Cu)). Microcapsules were filtered, washed with deionized water and phosphate saline buffer, and stored in deionized water at 4 °C until further studies. A number of the microcapsules were allowed to air-dry at room temperature to reach their equilibrium moisture content.

### 3.4. Evaluation of Cu(II) Complex Plant Growth-Regulating Activity

#### 3.4.1. Testing of Plant Growth-Regulating Activity

Seeds of corn (line P-346), barley (sort Zernogradsky), winter wheat (sort Dneprovsky 846) were germinated in aqueous solutions of Cu(II) complex. The seeds grown or soaked in water were as a control. A 2.5% sodium humate solution was used as a standard.

During germination, the following thermal regimes were maintained: for corn 25 °C, barley 20 °C, winter wheat 20 °C. Seeds of the test cultures were grown on aqueous solutions of the tested preparations (0.01, 0.001, 0.0001%) in the devices (seedlings) on filter paper. 50 pcs of barley and winter wheat seeds and 30 pcs of corn seeds were each placed in a germinator. Before seed hatching, the seedlings were covered with dark paper, and then germinated during the photoperiod night: day 8:16 and temperature 22–24 °C. Plant analysis was performed on the 14th day. The repetition of the experiment is four-fold.

Seed cultivation was carried out in 500 mL chemical glasses with agar-agar. Initially, a minimal amount of water was poured into the glass to avoid dilution, and then water was added to the initial mark. For the first six days, the seeds were in complete darkness at a temperature of 22 °C. Then the growth took place at a photoperiod of 6:18. Biometric analysis of plants was carried out on day 14.

#### 3.4.2. Testing of Auxin- and Gibberellin-Like Activity

The method of direct growth of segments of coleoptiles of etiolated seedlings with the fertilized top was used for auxin-like activity determination. The optimal test for auxin-like activity testing is corn coleoptiles due to their large size. The coleoptile of cereal plants is very sensitive to exogenous indoleacetic acid (IAA), and therefore it is a classic object for biological tests for auxin. For example, it is known that the growth of oat coleoptiles correlates well with the level of diffuse indoleacetic acid [48].

As a highly specific test for gibberellin, cuts of four-day-old corn seedlings with a coleoptile knot are used [49]. These segments were placed in Petri dishes with the test solutions and incubated for 48 or 72 h in a dark thermostat at 26 °C. The gibberellic acid solution was used as a standard. The growth of the first leaf protruding above the cylinder of the coleoptile was measured.

All measurements were repeated three times, and the results are presented as the mean with the corresponding standard deviation.

### 3.5. Fourier Transform Infrared Spectroscopy Analysis (FTIR)

The Fourier transform infrared spectroscopy (FTIR) spectra were recorded with the FTIR Instrument—Cary 660 FTIR (MIR system) spectrometer (Agilent Technologies, USA). Single samples of alginate, calcium chloride, chitosan, and Cu(II) complex, and their mixtures or microcapsules. Spectral scanning was done in the range of 400–4000 cm^−1^.

### 3.6. Microcapsules Physicochemical Characterization

#### 3.6.1. Differential Scanning Calorimetry (DSC)

DSC scans of Cu(II) complex, CS/(ALG/Ca), and CS/(ALG/Ca+Cu) were determined by differential scanning calorimetry (DSC) (Perkin Elmer DSC 6000). Approximately 5 mg of the sample was weighed into an aluminum pan which was crimped non-hermetically and heated. The heat evolved during the heating process (10°C min^−1^) from 20 °C to 350 °C was recorded as a function of temperature. DSC measurements were made at nitrogen atmosphere (N_2_ flow of 100 mL/min). Two measurements per sample were made and subsequent processing of the obtained results was performed using Pyris v11.0 software. Before analysis, temperature and heat flow calibration was performed using zinc and indium as reference materials.

#### 3.6.2. Microscopic Observations

Microcapsules size, morphology, and topography were analyzed by microscopic techniques: (i) optical microscopy (OM) (Leica MZ16a stereo-microscope, Leica Microsystems Ltd., Switzerland), (ii) scanning electron microscopy (SEM) (FE-SEM, model JSM-7000 F, Jeol Ltd., Japan) and atomic force microscopy (AFM) (Bruker Billerica, USA).

The average diameter of wet and dry microparticles was determined by optical microscopy using Olympus Soft Imaging Solutions GmbH, version E_LCmicro_09Okt2009. Twenty microparticles were randomly selected from batches produced in triplicate, to determine the size distribution.

Dry microcapsules formulations for SEM analysis were put on the high-conductive graphite tape. FE-SEM was linked to an EDS/INCA 350 (energy dispersive X-ray analyzer) manufactured by Oxford Instruments Ltd. (UK).

The samples for AFM imaging on air were prepared by the deposition of a microparticle suspension on the mica substrate. The microparticles are flushed three times with 50 μL of MiliQ water to remove all residual impurities. The microparticle surface topology, grain size distribution and cross-section- and roughness analysis within each sample were analyzed using MultiMode Scanning Probe Microscope with Nanoscope IIIa controller (Bruker, Billerica, USA) with SJV-JV-130 V (“J” scanner with vertical engagement); Vertical engagement (JV) 125 μm scanner (Bruker Instruments, Inc.); Tapping mode silicon tips (R-TESPA, Bruker, Nom. Freq. 300 kHz, Nom. spring constant of 40 N/m). In this manner, three-dimensional information about the surface topology was obtained [25]. All AFM imaging was performed at three different regions of each microparticle to ensure consistency of the obtained results.

### 3.7. Encapsulation Efficiency, Loading Capacity, and Swelling Degree

#### 3.7.1. Encapsulation Efficiency

The encapsulation efficiency (EE) was calculated from the total concentration of Cu(II) complex (*c_tot_*) and content of Cu(II) complex in dry microcapsules (*c_load_*) by the method of Xue et al. (2004) [50,51]. Encapsulation efficiency is calculated by the equation:EE = (𝑐_𝑙𝑜𝑎𝑑_/𝑐_𝑡𝑜𝑡_) × 100(2)
where 𝑐_𝑙𝑜𝑎𝑑_ = 𝑐_𝑡𝑜𝑡_ − 𝑐_𝑓_ and 𝑐_𝑓_ is the concentration of the Cu(II) complex in the filtrate.

The solution was vortexed again and left in the dark for 15 min followed by a reading of the value on a spectrophotometer at λ = 380 nm.

#### 3.7.2. Loading Capacity

The microcapsules loaded with Cu(II) complex were air-dried at room temperature for several days until reaching a constant weight. The Cu(II) complex content in a microcapsule was determined by dissolving 0.5 g microcapsules in 25 mL of a mixture of 16.80 g (0.2 mol dm^−3^) NaHCO_3_ and 17.65 g (0.06 mol dm^−3^) Na_3_C_6_H_5_O_7_ · 2H_2_O at pH 8. A mixture of buffer and microcapsules was placed on a magnetic stirrer (IKA Topolino) at 400 rpm until all microcapsules have completely dissolved. The obtained solution is filtered and the concentration of Cu(II) complex in the filtrate was determined by UV-VIS spectrophotometer (Shimadzu, UV-1700, Japan). The loading capacity (LC) is expressed as an amount of Cu(II) complex in mg per 1 g of dry microcapsules and calculated by the equation:𝐿𝐶_Cu(II)_ = (𝑤_Cu(II)_ /𝑤*_m_*_𝑐_)(3)
where *w_Cu(II)_* is the weight of the loaded Cu(II) complex in the sample and *w_mc_* is the weight of the microcapsules.

#### 3.7.3. Swelling Degree

Swelling depends, among other things, on the property of the dissolving agent, therefore, to avoid the influence of electrolytes from buffer solutions, the degree of swelling (Sw) was determined for microcapsules dispersed in deionized water. Microcapsules (10 mg) were transferred into tubes allowed to swell at room temperature for three hours in 10 mL of distilled water to strike a balance. The weight of the moist swollen microcapsules was determined by weighing after absorbing moisture from the surface of the microcapsule using filter paper. Swelling degree (Sw) was calculated using the equation:𝑆𝑤/% = ((𝑤_𝑡_ − 𝑤_0_)/𝑤_0_)(4)
where *w*_t_ is the weight of the swollen microcapsules and *w*_0_ is the weight of the dry ones. All measurements were repeated three times, and the results are presented as the mean with the corresponding standard deviation.

### 3.8. In Vitro Cu(II) Complex Release Profile from Microcapsules Formulations

In vitro studies of Cu(II) complex release from CS/(ALG/(Ca+Cu)) were investigated by dispersing microcapsules (30 g) in 100 mL of deionized water and allowed to stand during the experiment, without stirring, at room temperature. At appropriate time intervals, the dispersion was stirred for 60 s, an aliquot was taken and the concentration of Cu(II) complex was determined spectrophotometrically at λ = 380 nm.

### 3.9. Statistical Analysis

Microcapsules characterization experiments were carried out at room temperature in triplicate. The obtained data were analyzed with Microsoft Excel 2016 and all data were shown as a mean value with standard deviation.

## 4. Conclusions

Novel plant growth regulator, trans-diaqua-trans-bis [1-hydroxy-1,2-di (methoxycarbonyl) ethenato] copper (abbreviated as Cu(II) complex) was synthesized and its growth-regulating activity on several plant species from Ukraine was examined. Chitosan-coated calcium alginate microcapsules loaded with Cu(II) complex were prepared and characterized to be used as a device for controlled delivery to the plants.

The Cu(II) complex is a source of microelement, copper, which has a triple effect on plants, nourishes, protects, and regulates growth. An additional advantage is that by encapsulating in alginate microcapsules with calcium as a gelling cation, we supply the plants with an important macroelement, calcium. The growth-regulating properties of Cu(II) complex depend on the type of culture. The use of the Cu(II) complex accelerated the development of the corn root system and plant stems, but a lesser effect was observed on the development of wheat and barley root systems.

The structure of microcapsules and, consequently, the physicochemical characteristics depend on the molecular interactions between microcapsule constituents and Cu(II) complex, identified as electrostatic interactions and hydrogen bonds. FTIR and DSC analysis confirmed compatibility among all microcapsule formulation constituents.

Due to the presence of two gelling cations in the alginate matrix, the ionic crosslinks between different alginate chains are relatively strong forming a denser network. Both, CS/(ALG/Ca) and CS/(ALG/Ca+Cu)) microcapsules exhibit a granular surface structure with substructures consisting of fine substructural nanoparticle ordering. By loading the Cu(II) complex, the surface of the microcapsule becomes smoother and the granular surface with reduced roughness is less pronounced compared to the CS/(ALG/Ca) microcapsules. The release profile of Cu(II) complex from microcapsule formulations exhibited rapid initial release followed by a slower release obeying the power-law equation. Fickian diffusion was detected as the release rate-controlling mechanism.

Obtained results show that Cu(II) complex loaded in biopolymer microcapsules are suitable for application as enhanced fertilizer with the potential to be used in agricultural production of plant cultures. The benefits of new microcapsule formulations are the protection of encapsulated Cu(II) complex from degradation and reduced amount used for application as well as a controlled release for plant nutrition, protection, and growth regulation. Our future investigations are directed to in vivo application of new microcapsule formulations in an open field.

## Figures and Tables

**Figure 1 ijms-22-02663-f001:**
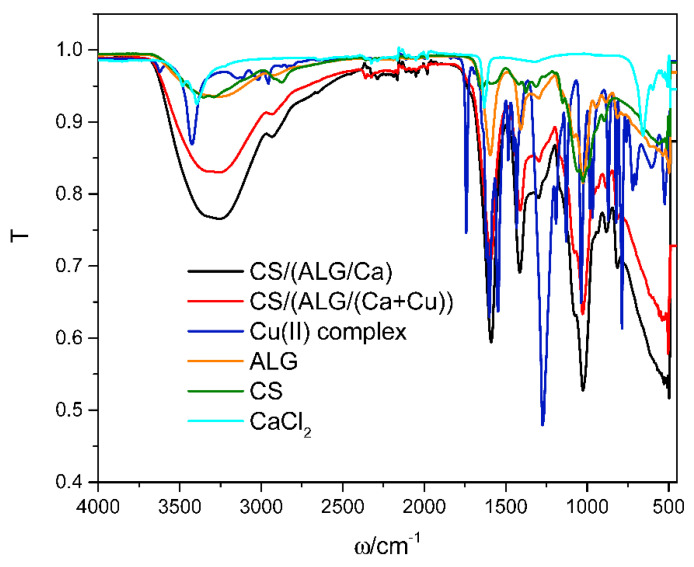
Characteristic FTIR spectrum of dry CS/(ALG/Ca) (black line) and CS/(ALG/(Ca+Cu)) (red line) microcapsules. A spectrum of single components, Cu(II) complex (blue line), ALG (orange line), CS (olive line), and CaCl_2_ (cyan line) are added for comparison.

**Figure 2 ijms-22-02663-f002:**
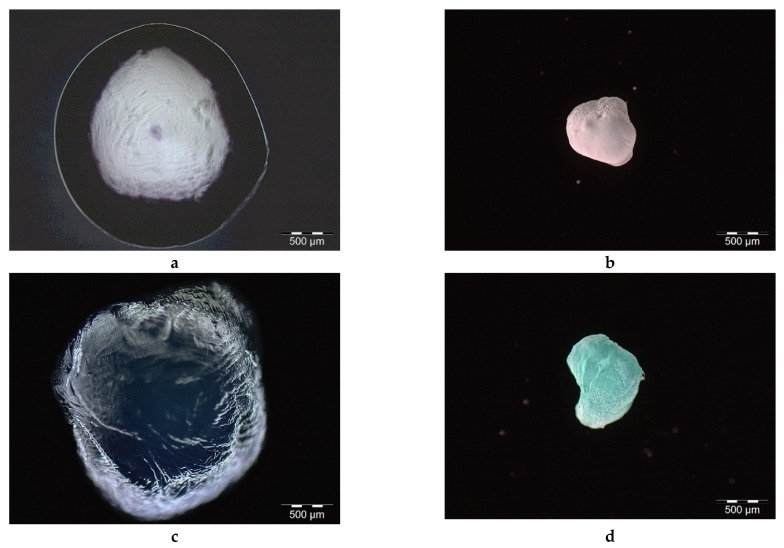
Optical microscope images of wet (**a**) and dried (**b**) CS/(ALG+Ca), and wet (**c**) and dried (**d**) CS/(ALG/(Ca+Cu)) microcapsules. Bars are indicated.

**Figure 3 ijms-22-02663-f003:**
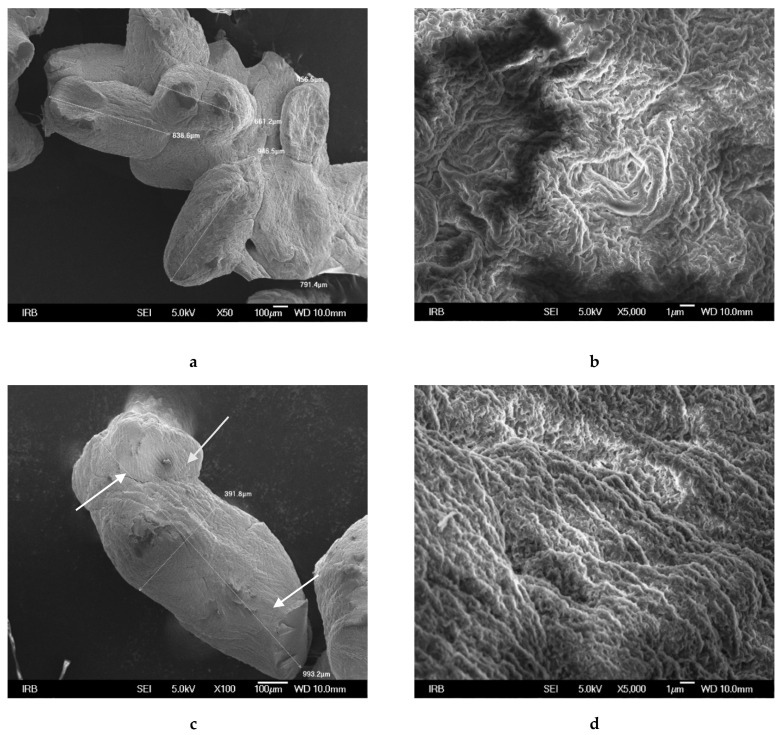
Scanning electron microscope images of CS/(ALG/Ca) (**a**,**b**) and CS/(ALG/Ca+Cu)) (**c**,**d**) dried microcapsules. Bars are indicated.

**Figure 4 ijms-22-02663-f004:**
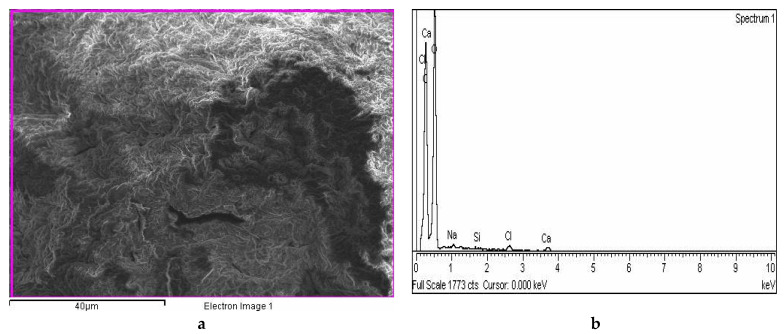
Surface morphology of CS/(ALG/Ca) (**a**) and CS/(ALG/(Ca+Cu)) (**c**) microcapsules, and surface elemental analysis (%) of CS/(ALG/Ca) (**b**) and CS/(ALG/(Ca+Cu)) (**d**) using dispersive X-ray spectroscopy (EDX). Bars are indicated.

**Figure 5 ijms-22-02663-f005:**
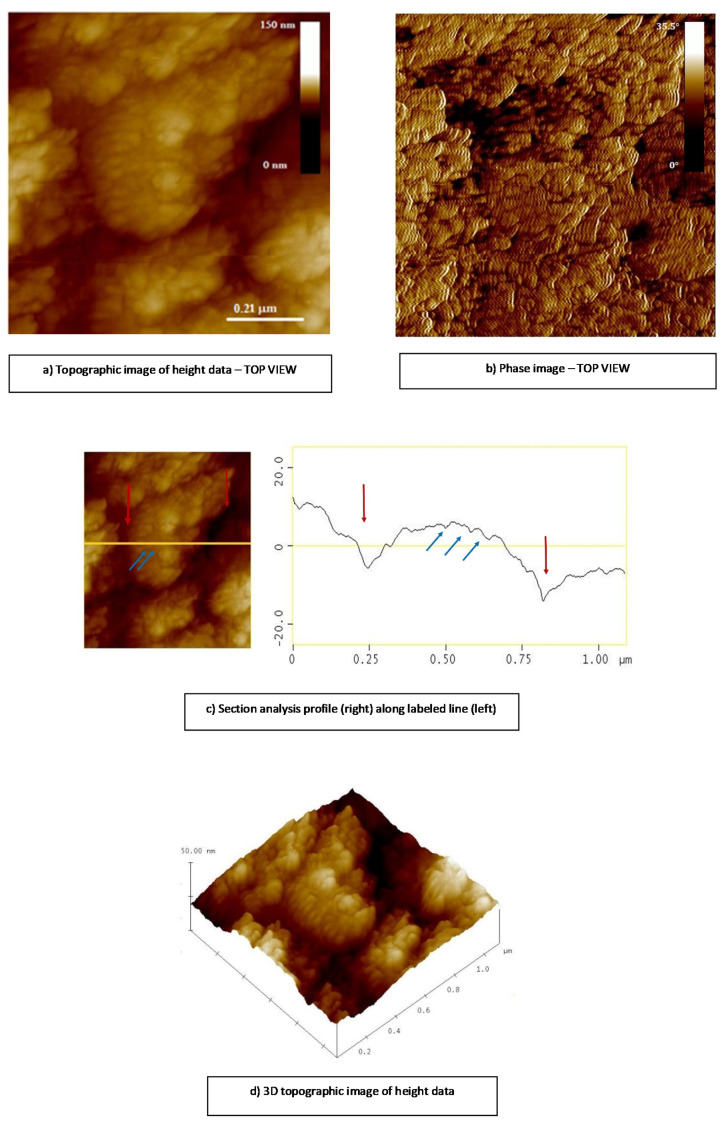
Top view of 2D-height data (**a**) 2D—phase data (**b**). A cross-section profile (**c**) along the yellow line (left) corresponds to the surface characteristics and 3D—height data (**d**) of CS/(ALG/(Ca+Cu)) microcapsules.

**Figure 6 ijms-22-02663-f006:**
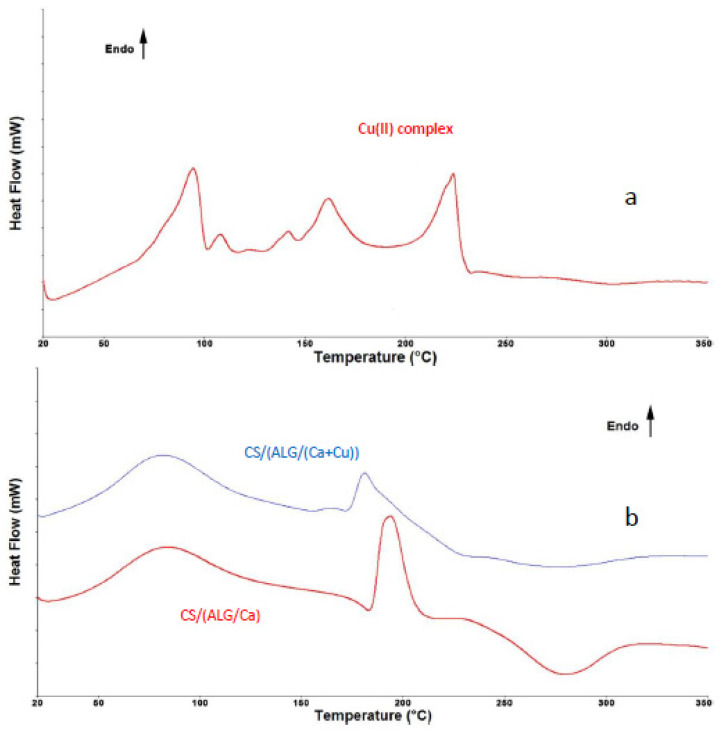
DSC thermograms of (**a**) Cu(II) complex and (**b**) CS/(ALG/Ca) and CS/(ALG/(Ca+Cu)) microcapsules determined under N_2_ flow.

**Figure 7 ijms-22-02663-f007:**
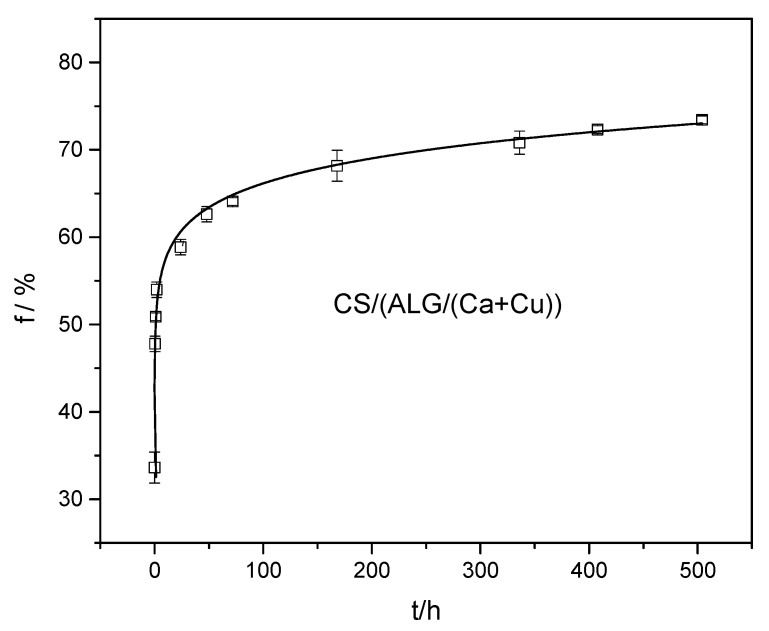
Percentage of released Cu(II) complex, (*f*) with time from CS/(ALG/(Ca+Cu)) microcapsules.

**Table 1 ijms-22-02663-t001:** Growth-regulating activity of trans-diaqua-trans-bis [1-hydroxy-1,2-di (methoxycarbonyl) ethenato] copper (*n* = 30). The seeds grown or soaked in water were as a control. A 2.5% sodium humate solution was used as a standard. ***c*** denotes concentration. Values of the deviation probabilities of the null hypothesis P, obtained from the data for Cu (II) in the concentrations 0.01, 0.001, and 0.0001%, Na humate 2.5% to the control (water).

		The Length (Weight) of Parts of Plants, % to Control			
Variant	*c* [%]	Root		Stem	
**Corn**					
**Cu (II)**	0.01	84.7 ± 0.6	(104.6 ± 0.3)	84.8 ± 0.5	(101.3 ± 0.3)
	0.001	129.7 ± 0.8	(144.4 ± 0.4)	107.6 ± 0.7	(141.0 ± 0.3)
	0.0001	146.0 ± 0.7	(155.5 ± 0.8)	128.4 ± 0.7	(121.8 ± 0.6)
**Na humate**	2.5%	103.1 ± 0.5	(95.5 ± 0.4)	101.4 ± 0.6	(96.2 ± 0.5)
**Barley**					
**Cu (II)**	0.01	96.4 ± 0.6	(101.8 ± 0.4)	104.6 ± 0.3	(106.6 ± 0.4)
	0.001	91.8 ± 0.5	(90.7 ± 0.5)	98.7 ± 0.6	(102.2 ± 0.4)
	0.0001	104.9 ± 0.4	(98.1 ± 0.3)	100 ± 0.7	(104.3 ± 0.6)
**Na humate**	2.5	96.8 ± 0.6	(90.9 ± 0.2)	101.6 ± 0.5	(96.0 ± 0.6)
**Wheat**					
**Cu (II)**	0.01	107.4 ± 0.7	(112.1 ± 0.6)	93.1 ± 0.6	(90.4 ± 0.5)
	0.001	103.1 ± 0.8	(106.9 ± 0.7)	97.9 ± 0.6	(97.4 ± 0.7)
	0.0001	106.7 ± 0.9	(108.6 ± 0.7)	96.6 ± 0.6	(94.7 ± 0.5)
**Na humate**	2.5	97.9 ± 0.9	−(93.1 ± 0.8)	101.4 ± 0.4	−(93.9 ± 0.7)
		**The Length (Weight) of Parts of Plants (Statistical analysis)**			
**Variant**		**Root**		**Stem**	
**Corn**		1.46702 × 10^−16^	5.3664 × 10^−18^	8.0298 × 10^−15^	3.84893 × 10^−17^
**Barley**		2.2209 × 10^−10^	1.91854 × 10^−12^	4.07004 × 10^−7^	7.04752 × 10^−10^
**Wheat**		7.28339 × 10^−8^	5.17777 × 10^−11^	1.70924 × 10^−8^	1.22409 × 10^−8^

**Table 2 ijms-22-02663-t002:** Influence of Cu(II) complex solutions of different concentrations (*c*) on the growth of coleoptile segments (mm) in 24 h (*n* = 45) *. Water was used as a control and indole-3-acetic acid (IAA) as a standard.

		Growth	
Variant	*c* [mg L^−1^]	mm	%
**Corn**			
**Cu (II)**	0.1	3.70 ± 0.05	115 ± 1.35
	1	3.50 ± 0.06	109 ± 1.71
	10	3.38 ± 0.05	105 ± 1.48
**IAA**	5	7.27 ± 0.03	226 ± 0.41
**H_2_O**	-	3.21 ± 0.06	100 ± 1.86

Note *—hereinafter means the number of cuttings or plants used in the experiment.

**Table 3 ijms-22-02663-t003:** Influence of Cu(II) complex solution (a) on the growth of whole corn coleoptiles in 24 h and (b) on the growth of the 1st leaf of the sections of corn seedlings in 24 h ^a^ (*n* = 30). Water was used as a control, indole-3-acetic acid (IAA) and gibberellic acid (GA) as standards.

		Growth	
Variant	*c* [mg L^−1^]	mm	%
**Corn**			
**(a)**			
**Cu (II)**	0.1	3.15 ± 0.06	94 ± 1.90
**IAA**	5	4.45 ± 0.04	133 ± 0.90
**H_2_O**	-	3.35 ± 0.06	100 ± 1.79
**(b)**			
**Cu (II)**	0.1	3.9 ± 0.05	91 ± 1.28
**GA**	10	6.2 ± 0.03	144 ± 0.48
**H_2_O**	-	4.3 ± 0.05	100 ± 1.16

^a^ measurements were taken from the base of the coleoptile.

**Table 4 ijms-22-02663-t004:** Peak transition temperatures (T_peak_) and enthalpy change (ΔH) of Cu(II) complex and CS/(ALG/Ca) and CS/(ALG/(Ca+Cu)) microcapsules determined under N_2_ flow.

Peak Position	Reaction	T_onset_/°C	T_end_/°C	T_peak_/°C	ΔH/Jg^−1^
Cu(II) complex					
1.	Endo	79.6	100.4	94.6	59.40
2.	Endo	102.5	113.4	108.1	5.23
3.	Endo	131.9	145.1	141.4	4.79
4.	Endo	151.8	173.8	161.8	32.91
5.	Endo	210.7	228.2	223.8	54.78
CS/(ALG/Ca)					
1.	Endo	42.0	119.8	80.4	113.03
2.	Endo	186.0	205.4	195.0	104.68
3.	Exo	240.8	308.5	278.8	128.64
CS/(ALG/(Ca+ Cu))					
1.	Endo	50.2	115.3	81.0	75.06
2.	Endo	175.7	189.9	181.0	100.96
3.	Exo	245.3	314.8	276.0	26.33

**Table 5 ijms-22-02663-t005:** Swelling degree (S_w_) of CS/(ALG/Ca) and CS/(ALG/(Ca+Cu)) microcapsules.

Sample	CS/(ALG/Ca)	CS/(ALG/(Ca+Cu))
S_w_	91.4 ± 3.4	53.8 ± 0.9

**Table 6 ijms-22-02663-t006:** Values of the constant (*k*), exponent (*n*), and correlation coefficient (*R*^2^) of Cu(II) complex released from (CS/(ALG/(Ca+Cu)) microcapsule.

Microcapsule Formulations	a	*k*/h	*n*	*R* ^2^
CS/(ALG/(Ca+Cu))	44.3 ± 3.0	8.2 ± 0.2	0.20	0.997

## Data Availability

The data presented in this study are available on request from the corresponding author.

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
