# Peer review of "Encapsulation of Synthesized Plant Growth Regulator Based on Copper(II) Complex in Chitosan/Alginate Microcapsules"

_ijms, 2021, doi:10.3390/ijms22052663_

Round 1

Reviewer 1 Report

Review - Manuscript ID: International Journal of Molecular Sciences-1110951

Encapsulation of synthesized plant growth regulator based on copper (II) complex in chitosan/alginate microcapsules

General remarks

The work presented addresses an interesting and topical subject. However, even if the main hypothesis of this study is that by encapsulating the Cu(II) complex the authors can prepare an improved agroformulation with copper delivery for the entire crop cycle, the fact remains that, in the absence of statistical analysis, the claimed physiological effects (plant-growth regulator) still deserve to be demonstrated.

Major remarks

Abstract: In the context of this abstract, the authors use a term (i.e. “significantly”) which seems to me totally inappropriate insofar as no real statistical approach is presented within the framework of this study to evaluate plant growth.

In general, the tables presented by the authors deserve to be improved in their form in order to make them more understandable by the reader. In addition, these must be completed statistically:

Table 1 (page 3):

  • The position of the variable C (%) should be reviewed in our opinion. Indeed, this variable corresponds to the columns of the different concentrations used and not to the different treatments applied.
  • Why do the authors not include certain values (noted “-“) corresponding to Cu (II) and the standard for wheat?
  • If the data in the table 1 correspond to averages, please show the corresponding values of SD.

Table 2 (page 3):

  • The same remark can be made for the term "Growth" which corresponds to the "mm" and "%" columns and not to the column for the different treatments. (Confers Table 3).
  • If the data in the table 2 correspond to averages, please show the corresponding values of SD.

Table 3 (page 4):

  • If the data in the table 3 correspond to averages, please show the corresponding values of SD.

Table 2 and 3: Why the authors limit themselves to presenting the results obtained from corn coleoptiles whereas if we refer to paragraph 3.4.2. of the document, all three species (barley, wheat and corn) were assessed?

Table 5 (page 12):

  • It seems that this table actually corresponds to table 4 and not to table 5 because table 4 does not exist upstream?
  • The term "Cu (II) complex" should be aligned like the other two treatments applied in this table.

Line 308-309: It seems that the elements mentioned in the context of these two lines (exothermic peak values) are not in agreement with the values presented in the corresponding table 5. Thus, if we take the authors' sentence: "Complete polymer decomposition is confirmed with the formation of exothermic peaks at 278.2 ° C and 276.0 ° C for CS/(ALG/Ca) and CS/(ALG/Ca + Cu) samples, respectively [36,37] ". However, in Table 5 these values are reversed: 276 ° C for CS/(ALG/Ca) and 278 ° C for CS/(ALG/Ca + Cu). So, thanks to the authors to check this and correct lines 310-312 accordingly.

Minors remarks

We advise authors to use the units conventionally used throughout the text in accordance with editorial rules; for example, "mL" instead of "ml" ...

Figure 1 and 7: The units of the different variables used in these figures should appear in brackets.

Line 383: Thanks to the authors for specifying the units of molecular weights.

Line 423: I propose to add the qualifier "thermal" to the term "regime" here.

Line 436: In Table 1 and in paragraph 3.4.1, seeds of maize, wheat and barley are mentioned. However, here, the authors evoke oat coleoptiles? Is this a mistake?

Line 525: Thank you to the authors to specify if any statistical analysis of the physiological results was made because paragraph 3.9 is not spoken?

Lines 562-570: To identify the contributions of the different authors, please use their initials (initials used for the title page).

Author Response

Dear Sir,

Enclosed please find the revised Manuscript ID: ijms-1110951. We would like to express our thanks to the reviewers for helpful comments and suggested alterations which improved the quality of the paper. We have done all the requested alternations in the manuscript according to their comments and suggestions.  Following is the list of the changes made in the manuscript (inserted text in the red and deleted text in blue).

General remarks

Auxins (including IAA), cytokinins, gibberellins, etc. are phytohormones that control plant development. As the reviewer correctly notes, plant growth regulators are organic substances of natural or synthetic origin, which can not only stimulate plant growth and development but also inhibit and slow down these processes. But synthetic origin means not only the resynthesis of known phytohormones but also the synthesis of any organic compounds that are not phytohormones but have growth-regulating abilities. There are a lot of such compounds. These include not only compounds that promote plant development, for example, para-aminobenzoic acid, α-naphthylacetic acid, CCC, mepiquat chlorix PIX, 4-chlorophenoxyacetic acid, and thousands of others, but also desiccants such as "the famous» Roundup. Therefore, any organic compound capable of influencing plant development is a plant growth regulator, including the above complex with the Cu (II) cation.

As an organic component, the complex contains the enol form of oxaloacetic acid, which is part of the Krebs tricarboxylic acid cycle. In turn, in the Krebs cycle, almost all metabolic pathways for the conversion of carbohydrates and amino acids in all living organisms are realized. Accordingly, the complex under consideration has been created on a natural organic compound that plays an important role in the metabolism of substances in any biological cell.

The considered complex was first obtained by the authors of the article. It has, as can be seen from the data in Table 1, growth-regulating properties. This is the first work to provide some data on these properties. Therefore, naturally, this compound cannot be known in scientific circles as a plant growth regulator. For the first time in this article, we declare this.

Indeed, copper is a trace element that is essential for plant development. 1 kg of fertile soil should contain about 5 mg of copper. Copper salts are indeed used as fungicides. It's all about the dose. A cobra bite can kill a person, but in small doses it is a medicine. The use of the complex under consideration in the concentrations indicated in the article suggests that the doses of copper cations at using the complex as a plant development stimulator will be no more than 1 g per hectare, that is, even much less than this microelement is required for normal plant development. Moreover, copper is entered as a component of the organic compound, which facilitates its entry into the plant.

Line 4 – inserted “Kudasova Darikha2

Line 10 – inserted „(B.Z.M.); dariha_uko@mail.ru (K.D.)

Reviewer 1

Major remarks:

Line 22 – deleted „significantly“

Lines 88 to 95 – inserted “Any organic compound capable of influencing plant development is a plant growth regulator, including the above complex with the Cu(II) cation. As an organic component, the complex contains the enol form of oxaloacetic acid, which is part of the Krebs tricarboxylic acid cycle. In turn, in the Krebs cycle, almost all metabolic pathways for the conversion of carbohydrates and amino acids in all living organisms are realized. Accordingly, the complex under consideration has been created on a natural organic compound that plays an important role in the metabolism of substances in any biological cell. The considered complex was first obtained by the authors of the article.“

Line 119 – inserted „c denotes concentration“

Tables 1, 2, 3 – improved in their form and statistically completed according to suggestions of both reviewers.

Line 322 – inserted „4“ instead of „5“

Line 339 – inserted „4“ instead of „5“

Minor remarks:

Conventional units are used throughout the text in accordance with editorial rules

Figure 1 and 7: The units of the different variables are used in these figures.

Line 353 – inserted „5“ instead „6“

Line 399 – inserted „6“ instead „7“

Line 408 – inserted „Da“

Line 449 – inserted „thermal“

Line 442 – delete „oats“

Line 571 to 580 – authors' initials are used.

Material and Methods - This part was improved and some details are added.

Line 462, 463 – deleted „oats, wheat or corn“

Lines 463, 464 – inserted „The optimal test for auxin-like activity testing are corn coleoptiles due to their large size.“

Lines 466, 468 – inserted „For example, it is known that the growth of oat coleoptiles correlates well with the level of diffuse indoleacetic acid“ instead of „The growth rate of oat corn coleoptiles correlates well with the level of diffuse indoleacetic acid.“

Lines 4,75,476   - inserted „All measurements were repeated three times, and the results are presented as the mean with the corresponding standard deviation”.

Lines 656 to 673 – inserted references:

  1. Singh, A.; Chaudhary, A. Synthesis, Spectral Characterization and Biological Study of Heterobinuclear Complexes of Cu(II) with Si(IV). Silicon, 2019, 11, 1107–1118. https://doi.org/10.1007/s12633-018-9971-4
  2. Deosarkar, S.D.; Chavan, S.A.; Puyad, A.I. Study of plant growth regulating activity of (2-chlorophenyl) (5-(2 hydroxyphenyl)-3-(pyridin-3-yl)-1H-pyrazol-4-yl) methanone and itsFe (III) and Cu (II) complexes on Trigonella foenum-graecum. J. Chem. Pharm. Res., 2011, 3, 703-706. URL: http://jocpr.com/vol3-iss4-2011/JCPR-..
  3. Deosarkar, S.D. Plant growth regulating 4-methylbenzenesulfonic acid transition metal complexes. J. Chem. Pharm. Res. 2012, 4, 592-595. https://www.jocpr.com/archive/
  4. Lopez-Lima, D.; Mtz-Enriquez, A.I.; Carri´on , G.; Basurto-Cereceda, S.; Pariona, N. The bifunctional role of copper nanoparticles in tomato: Effective treatment for Fusarium wilt and plant growth promoter. Scientia Horticulturae 2021, 277, 109810. https://doi.org/10.1016/j.scienta.2020.109810.

  instead of

„13. Han, X.; Chen, S.; Hu, X. Controlled-release fertilizer encapsulated by starch/polyvinyl alcohol coating. Desalination 2009, 240, 21–26, doi:10.1016/j.desal.2008.01.047.

  1. Yuvaraj, M.; Subramanian, K.S. Controlled-release fertilizer of zinc encapsulated by a manganese hollow core shell. Soil Sci. Plant Nutr. 2015, 61, 319–326, doi:10.1080/00380768.2014.979327.
  2. Chen, C.; Gao, Z.; Qiu, X.; Hu, S. Enhancement of the Controlled-Release Properties of Chitosan Membranes by Crosslinking with Suberoyl Chloride. Molecules 2013, 18, 7239–7252, doi:10.3390/molecules18067239.
  3. Neelakantan, M.; Esakkiammal, M.; Mariappan, S.; Dharmaraja, J.; Jeyakumar, T. Synthesis, Characterization and Biocidal Activities of Some Schiff Base Metal Complexes. Indian J. Pharm. Sci. 2010, 72, 216, doi:10.4103/0250-474X.65015.
We hope that after suggested changes made in the manuscript and given explanations you will find the manuscript acceptable for publication in the International Journal of Molecular Sciences.

Sincerely,

Marko Vinceković

Reviewer 2 Report

The manuscript describes the preparation and characterization of microcapsules suitable for delivery of synthesized Cu(II) complex to the plants as a new agroformulation of a plant growth regulator (PGR). This is a relevant topic for development of new fertilizers or plant protection products.

However there are some weaknesses explained below:

In this work, the synthesized  copper compound is classified as a plant growth regulator (PGR). However PGRs are auxins, cytokinins, gibberellins, IAA, ethylene, salicylic acid, jasmonates, i.e., organic substances produced naturally by plants or synthesized.  

Copper is a micronutrient, that can be applied as a fertilizer, in very low dosis, in copper deficient soils. Its main use is for prevention of fungal diseases and is one of the main heavy metal soil polutant. It is also phytotoxic for plants as the authors observed in their experiments.

Please explain, and support, the PGR classification for Cu(II) complex, since, to my knowledge, this compound isn’t considered a PGR.

Lane 47-49 – Cu(II) complex is not a phytohormone, like other PGRs, therefore the sentence isn’t applied to the present study.

Lane 76-77 . This  isn’t supported by the references (Refs 13-16), because they were related with encapsulated controled-release fertlizers and not with the use of Cu(II) as plant growth-regulators. Please check the references.

Lane 90-92 - The main hypothesis of the work was to obtain an encapsulated Cu(II) complex, as a plant growth-regulator, for improved agroformulation for copper delivery. However, see above comments regarding classification of the copper micronutrient as a PGR.

Lane 101-116  - It was evaluated the effect of the compund on corn development, where the effect as a micronutrient was observed. Why authors conclude that observed a plant growth-regulating activity?

Lane 112 – development of corn system –should be corn root system

Table 2 Data is presented without statistic analysis and only the means (without standard deviation) are presented.

Lane 130-131- Only one concentration of IAA was tested, while 3 different concentrations were tested with Cu(II) on corn coleoptiles. Why authors conclude that the Cu (II) complex showed an auxin-stimulating effect on the growth of coleoptile segments, which decreased with increasing concentration? Why not a phytotoxic effect?

Lane 135 – Object?

Lane 141 – test object?

Please revise scale bar in Figure 4c

Material and Methods: Some details are missing and it was not easy to read. No statistical analysis was performed. Mean and standard deviation are described only for microcapsules characterization experiments but significance of the results are not presented.

Overall, the manuscript requires significant improvements.

Author Response

Dear Sir,

Enclosed please find the revised Manuscript ID: ijms-1110951. We would like to express our thanks to the reviewers for helpful comments and suggested alterations which improved the quality of the paper. We have done all the requested alternations in the manuscript according to their comments and suggestions.  Following is the list of the changes made in the manuscript (inserted text in red and deleted text in blue).

General remarks

Auxins (including IAA), cytokinins, gibberellins, etc. are phytohormones that control plant development. As the reviewer correctly notes, plant growth regulators are organic substances of natural or synthetic origin, which can not only stimulate plant growth and development, but also inhibit and slow down these processes. But synthetic origin means not only the resynthesis of known phytohormones, but also the synthesis of any organic compounds that are not phytohormones, but have growth-regulating abilities. There are a lot of such compounds. These include not only compounds that promote plant development, for example, para-aminobenzoic acid, α-naphthylacetic acid, CCC, mepiquat chlorix PIX, 4-chlorophenoxyacetic acid, and thousands of others, but also desiccants such as "the famous» Roundup. Therefore, any organic compound capable of influencing plant development is a plant growth regulator, including the above complex with the Cu (II) cation.

As an organic component, the complex contains the enol form of oxaloacetic acid, which is part of the Krebs tricarboxylic acid cycle. In turn, in the Krebs cycle, almost all metabolic pathways for the conversion of carbohydrates and amino acids in all living organisms are realized. Accordingly, the complex under consideration has been created on a natural organic compound that plays an important role in the metabolism of substances in any biological cell.

The considered complex was first obtained by the authors of the article. It has, as can be seen from the data in Table 1, growth-regulating properties. This is the first work to provide some data on these properties. Therefore, naturally, this compound cannot be known in scientific circles as a plant growth regulator. For the first time in this article, we declare this.

Indeed, copper is a trace element that is essential for plant development. 1 kg of fertile soil should contain about 5 mg of copper. Copper salts are indeed used as fungicides. It's all about the dose. A cobra bite can kill a person, but in small doses it is a medicine. The use of the complex under consideration in the concentrations indicated in the article suggests that the doses of copper cations at using the complex as a plant development stimulator will be no more than 1 g per hectare, that is, even much less than this microelement is required for normal plant development. Moreover, copper is entered as a component of the organic compound, which facilitates its entry into the plant.

Line 4 – inserted “Kudasova Darikha2

Line 10 – inserted „(B.Z.M.); dariha_uko@mail.ru (K.D.)

Reviewer 2

Line 49 – inserted „natural PGRs“ instead of „phytohormones“

Tables 1, 2, 3 – corrected according to suggestions of both reviewers

Line 100, 101 – inserted „a source of essential trace elements and a plant growth-regulator“ instead of „as a plant growth-regulator and a source of essential trace elements“.

Lines 90, 92 and 101 to 116 – This was explained in general remarks.

Lines 122 – inserted „root“.

Tables 1, 2, 3 – corrected according to suggestions of both reviewers.

Table 2 – Copper complex testing on segments of coleoptiles showed that it does not have an effect similar to auxins and gibberellins. For this reason, there was no need to investigate different concentrations of IAA.

Line 146 – inserted „corn“ instead of „this object“

Line 146 – inserted „only a mild“ instead of „an“

Line 147, 148 – inserted „almost disappeared“ instead of „decreased“

Line 172 to 179 – inserted „Biological testing of Cu (II) complex on segments of coleoptiles showed that it does not have an effect similar to auxin or gibberellin. The growth-promoting effects presented in Table 1 appear to be due to a different mechanism of action of the complex on plant development.” instead of “A biological test of auxin-like activity in 24 hours revealed only a mild auxin-stimulating effect on the growth of the corn coleoptile segment, and even a slight inhibition of the growth of the whole corn coleoptile at a concentration of 0.1 mg l-1. Compared to the controls gibberellin-like activity in 24 hours exhibited only a minor inhibition of the growth of the 1st leaf of parts of corn seedlings.”

Figure 4 - scale bar in Figure 4c is revised.

We hope that after suggested changes made in the manuscript and given explanations you will find the manuscript acceptable for publication in the International Journal of Molecular Sciences.

Sincerely,

Marko Vinceković

Round 2

Reviewer 2 Report

The manuscript was improved in the present version. However statistical analysis is still missing. Tables were completed with the standard deviation values but the analysis of variance wasn’t performed in order to determine the statistical differences and the significance of the results.

Please indicate the "n" in table 1.

In material and methods (3.4)  there is no information about the number of samples per treatments.

Author Response

Dear Sir,

Enclosed please find the revised Manuscript ID: ijms-1110951. We have done the requested alternations in the manuscript according to the comments.

Following is the list of the changes made in the manuscript.

Lines 123 to 125 – inserted „Values ​​of the deviation probabilities of the null hypothesis P, obtained from the data for Cu (II) in the concentrations 0.01, 0.001 and 0.0001% (only data with P <0.05 were taken into account) are presented in Table 1b.”

Line 128 – inserted „(n=30)“ in Table 1a.

Lines 132-135 – inserted „Table 1b“.

Line 466 – 3.4.1. – information on the number of samples is mentioned.

Lines 486, 487 – 3.4.2. – information on the number of samples is mentioned.

We hope that after suggested changes made in the manuscript you will find the manuscript acceptable for publication in the International Journal of Molecular Sciences.

Sincerely,

Marko Vinceković